# Towards Fully Autonomous Driving with Automated Commonsense Reasoning

## Abstract

Autonomous Vehicle (AV) technology has been heavily researched and sought after, yet there are no SAE Level 5 AVs available today in the marketplace. We contend that over-reliance on machine learning technology is the main reason. Use of automated commonsense reasoning technology, we believe, can help achieve SAE Level 5 autonomy. In this paper, we show how automated commonsense reasoning technology can be deployed in situations where not enough data is available to train a machine learning model for autonomous driving. Specifically, we consider two situations where (i) a traffic signal is malfunctioning at an intersection and (ii) all the cars ahead are slowing down and steering away due to an unexpected obstruction (e.g., animals on the road). We show that in such situations, our commonsense reasoning-based solution performs correctly. We also provide a pathway for efficiently invoking commonsense reasoning by measuring uncertainty in the computer vision model and using commonsense reasoning to handle uncertain scenarios. We describe our experiments conducted using the CARLA simulator and the results obtained. The main contribution of our research is to show that automated commonsense reasoning provides an effective pathway to reach SAE level 5 automation of autonomous driving.

## 1 Introduction

AV systems have made significant progress over the course of the past several years. Due to the nature of vehicles and road scenarios, it is highly critical that AV systems are perfectly accurate and can accommodate for uncertainties in driving situations such as obstacles, pedestrians, and objects not visible to the camera. These systems must also be able to adjust for road layouts and positions that may not have been encountered in training. A single mistake in classification can lead to accidents and casualties. However, an optimal AV system can assist or control many driving aspects, such as sensing, path planning, speed control, navigating, and parking, reducing accidents caused by human error (Parekh et al. (2022)). This ideal is difficult to achieve since AV systems are heavily dependent on deep learning technologies and may miss-classify due to uncertain variables on the road. Therefore, we believe that any improvement to the classification of these systems is significant to the development of safe and accurate AV vehicles.

One ideal for an AV system is to imagine it as a perfect human driver—one who is able to reason like a human who does not commit any errors in judgment or perception. This type of system can be considered as making use of Visual Question Answering (VQA) which gives answers to questions about the visual scenario around the vehicle. Such a system would be effective for both explaining decisions made by an autonomous vehicle to a human controller. However, neural network approaches to explaining visual scenarios tend to work by simply matching word descriptions to the data (Johnson et al. (2017)). Similar issues can be found in Large Language Models which can hallucinate, or generate irrelevant information, when answering questions. These systems heavily rely on matching a response to the query and lack true understanding of the context of the scenario. In a safety critical system, we must avoid making mistakes. We believe that commonsense reasoning can be used to properly model human thinking/reasoning (Gelfond & Kahl (2014); Kowalski (2011)).

We propose a system that will correct misclassification in the object detection phase of an autonomous vehicle model. The system consists of a base deep learning autonomous vehicle model with a logic program (Bratko (2012)) layer on top that is used to improve the output. Commonsense

reasoning, realized through logic programming, is used to detect potential inconsistencies in the classifications of the objects in the scenario. The logic program will then output the inconsistency and the explanation for why it believes there to be an issue. We then make adjustments to the output based on the suggested inconsistency to improve the overall accuracy of the computer vision model. In this paper, we will explore the class of possible improvements we can make by observing the collective behavior of vehicles around us to correct misclassification over traffic lights and obstacles in the road. One can model other complex situations that arise during driving in a similar way, resulting in nearly all misclassifications being detected and rectified, paving the way to SAE Level 5 autonomy.

The organization of our paper is as follows: Section 2 will cover background topics relating to AV systems, Commonsense reasoning and its relation to targeted AV scenarios, and deep learning techniques used to handle perception. Section 3 will motivate and contextualize our solution. Section 4 will describe our methodology and how we connect the logic to the perception for each of our targeted scenarios. Sections 5 will describe the experiments and evaluate the results. Section 6 covers recent related works in the field and will compare our approach to them. Finally, Section 7 will conclude, summarize the paper, and motivate our future work.

## 2 BACKGROUND

**Autonomous Vehicles:** Research in autonomous vehicles has experienced massive growth due to the popular development of deep learning and sensor research (Yurtsever et al. (2020)) causing an increase in advanced driver assistance systems (ADAS) in research and commercial vehicles (Van Brummelen et al. (2018)). However, there still exist major gaps in AV research that prevent it from achieving full automation, such as disagreements in optimal perception techniques, algorithms that lack efficiency and accuracy, and the inability to handle difficult road conditions or inclement weather (Yurtsever et al. (2020)).

Here we outline some of the formal definitions for autonomous vehicles that we will use to motivate the rest of the paper. SAE International defines the SAE Levels of Driving Automation into six overall levels (Committee (2021)) ranging from no driving automation to full driving automation, with levels 3-5 being defined as automated driving systems (ADS). These higher levels of automation have been heavily explored in recent AV research, though no SAE Level 5 vehicles have been realized. One of the issues preventing full automation is imperfect perception and sensing technologies, but another major issue lies in the higher-level aspects of driving.

Autonomous vehicle technology must deal with four major issues: (i) environment perception and modeling, (ii) localization and map building, (iii) path planning and decision-making, and (iv) motion control. We relate these aspects to the two types of systematic thinking—System 1 and System 2—proposed by Kahneman in 2011 (Kahneman (2011)). System 1 thinking is fast, intuitive, and reflexive, while System 2 thinking is slow and deliberative. Deep learning and computer vision techniques are a very effective and explored approach to handling the first two aspects of autonomous vehicles, as they are heavily based on pattern matching and System 1 thinking. However, the last two aspects require high level reasoning and are more in line with System 2 thinking. Many approaches to AV development consist solely of deep learning techniques for all four aspects, but there is a lot of potential in a hybrid solution that integrates vision and semantics (Suchan et al. (2020); Kothawade et al. (2021)). To this end, we motivate a hybrid approach using deep learning perception and commonsense reasoning. In this work, we will be simulating AV scenarios using the CARLA (Car Learning to Act) simulator (Dosovitskiy et al. (2017)). CARLA is an open platform that allows control over sensor suites and environmental factors.

**Commonsense Reasoning:** To achieve an ideal SAE level 5 autonomous vehicle, the system must be able to comprehend information and reason as a human would. This approach makes use of *commonsense reasoning* to emulate how a human would approach driving. This reasoning is performed over *commonsense knowledge* that we humans possess. Commonsense reasoning is able to handle uncertainties in our perception as well as fill in the gaps. Our approach relies on the observation that humans drivers use *perception* to detect objects (roads, pedestrians, other cars, curbs, traffic lights, lane markings, etc.) in the scene and then use (commonsense) *reasoning* to make driving decisions.

Commonsense reasoning is modeled with default rules with exceptions (Gelfond & Kahl (2014)). This represents the human way of thinking about the world around us. For example, a default rule about driving could be *"Drivers will typically stop at a red light"*. However, human knowledge will contain exceptions for those default rules, such as *"Irresponsible drivers don't always stop at red lights"*. This allows commonsense reasoning to handle uncertainty by creating a conclusion with given information and then updating that conclusion based on new incoming information.

The strength of commonsense knowledge and commonsense reasoning in an autonomous driving scenario is its ability to handle multiple scenarios without the need for retraining for each constrained scenario. We propose that commonsense reasoning can be layered on top of the computer vision model and make adjustments to the output based on detected inconsistencies between objects as new knowledge is observed in the world around an AV. For this experiment, commonsense knowledge is hand-coded as logic programs to build the knowledge base needed. The logic programs cover only the scenarios within our scope, specifically solving misclassifications of traffic lights and road obstacles using collective behavior. In the following sections, we give a high level description of how commonsense reasoning—that emulates human behaviour—can be applied to two different road scenarios.

## 3  MOTIVATION

While deep learning-based vision and control has seen success in AV technology, a hybrid system brings potential improvements in explainability, human-centred design, and compliance with human-defined industry standards (Suchan et al. (2021)). We motivate this paper by defining the differences between human behavior and commonplace AV behavior.

Deep learning is a powerful technology, especially for computer vision. However, due to external factors such as noise, camera or sensor degradation, inclement weather, unusual road scenarios, etc., can cause misclassifications in perception. Perception can also fail in training, and due to the rare nature of accidents, it is possible that an AV does not have sufficient experience with a specific scenario. Due to the black box nature of such systems, such faults in the training may only be noticed after the scenario is encountered, often leading to an accident. In a safety-critical system, such accidents are completely intolerable. It is better to create a set of human-defined commonsense rules that protect the AV when it gets into trouble due to such factors.

We take inspiration from the human approach to missing information on the road to write these rules. Like an AV system, it is possible for a human driver to misinterpret information on the road. However, for a human driver, it is possible to use the existing context to reason over the scenario and still make a safe decision. In this paper, we represent how humans look at nearby drivers to better understand a driving scenario. For example, a human driver understands that by default, vehicles are supposed to stop at a stop sign. However, if a human driver notices that a group of vehicles ahead aren't stopping for an incoming stop sign, then the stop sign was likely misinterpreted. This ability to reason over incorrect information allows human drivers to avoid potential accidents. We can use this commonsense reasoning approach to correct missing or incorrect information in the perception system to achieve 100% accuracy. This is because the commonsense rules are human-defined and can be continuously refined to completely represent an expert human driver.

**Traffic Lights:** The crux of the commonsense reasoning for these two scenarios that we consider is observation and reasoning over collective behavior. Our goal is to emulate what a human would do in a given scenario. If a human driver can't see what color an upcoming traffic light is, or if there is a traffic light at all, then they will observe the vehicles around them to come to a conclusion. Our commonsense reasoning module emulates human decision-making. If the vehicles traveling in the same direction as the ego vehicle are stopped, then our logic will say that there is a reason to stop; either a stop sign or a red traffic light is present ahead. If the model detects a collective movement from one of the lanes of traffic across the intersection, then the logic knows there is a traffic light at this intersection. Alternatively, if the model detects a collective "stop and go" pattern across the intersection, then the logic knows there is a stop sign.

A demonstration can be seen in figure 1. The perception, due to some noise or filter on the camera, initially detects the traffic lights in the image as being green. However, it also detects a group of vehicles on the other side of the intersection making a left turn. This information is logically

Figure 1: Examples of potential misclassifications for traffic lights and obstacles on the road. The left image shows a misclassified traffic light and the right image shows a misclassified road obstruction.

conflicting; however, the deep learning system on its own would allow this classification to exist. A human driver would use commonsense to reason that either the identified color of the traffic light is wrong or that the vehicles on the other side are committing a traffic violation. The latter is unlikely if it's a large group of vehicles. Because this scenario violates the default rules established by commonsense, a human-thinking emulating AV system should identify this and rectify the issue.

**Obstacles:** The logic for reasoning over potential obstacles is also based on collective behavior. The model detects cluster event changes, meaning it can tell when multiple vehicles change their action within one location. The logic will attempt to give a reason for this action change. For example, vehicles could be turning while we are near an intersection. If no reason can be given, then logic determines this to be an inconsistency, meaning that it is possible that vehicles are collectively changing their action due to an obstacle.

This is similar to how humans would determine potential obstacles while driving. For example, if there is some construction in the left lane and they don't happen to see it, they would know that something is odd if they suddenly see all the vehicles in front of them change to the rightmost lane when there is no intersection or turn nearby. In this scenario, even if they don't immediately see the construction, they would slow down to see what the problem is or naturally change into the rightmost lane, assuming that there is something blocking people from driving in the left lane.

In figure 1, we can see an example scenario where an obstruction on the road is not seen by the camera. This is an example of a scenario with missing or hidden information not caught by the perception model. In the image, a vehicle is blocking the obstruction from view. However, the camera does catch the vehicle and the vehicles before it makes a lane change to the left (a collective behavior). A human driver would notice the collective lane change and the slowdown of traffic and reason that there must be something blocking the right lane. An AV with commonsense knowledge should be able to generate the new knowledge that there is an obstruction there, even without the perception model being able to see it.

## 4 METHODOLOGY

Sensors commonly used in autonomous vehicles, such as cameras, LiDAR, radar, IMU, and GNSS, only produce low-level information, which is difficult to directly use in a commonsense reasoning system. In order to obtain high-level information that is more helpful to the models, we used several deep learning models.

Birds-eye-view (BEV) semantic segmentation is a cutting-edge technique widely used in autonomous driving. Given the image captured from surrounding cameras installed on an autonomous vehicle, BEV semantic segmentation model transforms those images into a top-down perspective and classifies them into predefined categories, such as roads, vehicles, pedestrians, and other objects. Compared to camera space semantic segmentation, the BEV semantic segmentation offers a clear and comprehensive view of the surroundings, enabling better decision-making processes.

In this paper, we consider Lift-splat-shoot (Philion & Fidler (2020)) as the BEV semantic segmentation model. The model takes images captured by six cameras installed around the vehicle as input and outputs a BEV semantic segmentation containing four classes: vehicles, drivable area, lane

makes, and others. In CARLA, we mounted six cameras on top of the ego vehicle at six different angles: -120, -60, 0, 60, 120, and 180. Each camera has a field-of-view (FOV) of 90 degrees. The birds-eye-view image is a $500 \times 500$ RGB image, representing a $100m \times 100m$ range, where the ego vehicle is located at the center of the image. The ground truth birds-eye-view image for training is created in Carla by mounting a camera 50 meters above the ego vehicle facing straight down, with a FOV of 90 degrees.

**Traffic Light Detection:** Traffic light information is very important for decision-making at intersections. Many traffic light poles contain multiple groups of traffic lights, each group corresponding to different directions and lanes. Depending on which lane our ego vehicle is in and which direction we want to go, we may need to look at different groups of lights. While detecting the state of the lights is relatively easy, understanding the meaning of the light can be challenging. In this paper, we consider a simpler setting; we only detect the traffic light that is impacting the ego vehicle. In other words, we're trying to figure out whether we should go or stop at the traffic light.

For this task, we trained a CNN that takes the front camera image to do a binary classification; the positive class means that the ego vehicle is being impacted by a red traffic light and thus it should stop, and the negative class means the vehicle is not being impacted by a red traffic light and it can go.

Carla provides an API *"Vehicle.get_traffic_light()"* which will return the state of the traffic light that is affecting the vehicle. When multiple vehicles are being affected by a traffic light, only the first vehicle is able to get the correct state, so we implemented a function to fetch the traffic light state of the first vehicle in a lane, and we used the return value of this function as the ground truth.

**Behavior Detection:** In order to detect collective behavior, we first need to detect the individual behavior of the vehicles. We defined 6 categories of events: *ChangeLaneLeft*, *ChangeLaneRight*, *Straight*, *TurnLeft*, *TurnRight*, and *LaneFollow*. In these six events, *Straight*, *TurnLeft*, and *TurnRight* can only happen at an intersection.

The behavior prediction model is a classification model based on "resnext50_32x4d". For each obstacle vehicle in the BEV range, we translate and rotate the BEV image so the obstacle vehicle is located at the center of the image and heading up. Then we predict the behavior of the vehicle.

Ground truth behavior is obtained by looking at the future trajectory of the vehicle. For example, if the vehicle is tuning right at an intersection, we mark the behavior of the vehicle to *TurnRight* from when it starts to turn until it finishes. With the knowledge of the future trajectory, we wrote a few simple pattern-matching rules to obtain the ground truth behavior. The model was then trained based on the collected ground truth data.

**Behavior Clustering:** Within a time window, if multiple vehicles close to each other perform the same behavior, we treat this group as a cluster. With the individual behavior known, we can detect behavior clustering with classical clustering algorithms. To obtain behavior clusters, we consider a 20-second sliding time window. In the time window, for each category of behaviors, we use DBSCAN on the location of where the behavior starts to try to form a cluster. DBSCAN eps is set to 2.7, and the minimum cluster size is set to 2. If a cluster is found, we save the information about the cluster for use in reasoning.

**Uncertainty Prediction:** The uncertainty of the deep learning models is evaluated with evidential deep learning (Sensoy et al. (2018)). Evidential deep learning estimates the aleatoric and epistemic uncertainty of an underlying model by replacing its output class probability with a Dirichlet distribution. We adopted the traffic light detection model and BEV semantic segmentation model and calculated both aleatoric and epistemic uncertainty. Traffic light detection uncertainty is a numerical value, and BEV uncertainty is obtained at pixel level.

**Commonsense Reasoning:** The main contribution proposed by this paper is using a commonsense reasoning layer to perform consistency checking over the deep learning model. The commonsense reasoning models commonly held information about how nearby vehicles behave when presented with certain traffic scenarios.

**Preprocessing for Commonsense Reasoning:** The deep learning model outputs a large amount of image and object data to represent each road scenario. For the dataset used in this experiment,

CARLA, we largely format the data in a similar manner so that the connection between the deep learning model and the logic program is the same. Each frame contains information about each object, such as vehicles, traffic lights, lanes, and intersections. A script is used to convert the data from these objects into facts that can be used by the commonsense reasoning module. The deep learning model will furthermore make predictions about the actions and the action change clusters of vehicles for each scenario. These predictions are also loaded as logic facts representing each object, which form the commonsense knowledge base. The results of these predictions are reasoned over, and if inconsistencies exist with the predictions, then the commonsense reasoning model will detect those inconsistencies.

**General Procedure:** Our approach has four major steps. In general, our goal is to represent perception information as knowledge, and give feedback to the AV by reasoning over that knowledge.

**Step 1:** Define commonsense reasoning rules to reason about the possible explanations for each specific scenario (e.g., traffic light, obstacle, stop signs).

**Step 2:** Perform commonsense reasoning about the relations between the collective behaviors of all vehicles and non-vehicle objects predicted by the autopilot deep learning model and identify the violated rules. Our experiments consider two approaches to this step: allowing the commonsense reasoning to be active the whole time the system is running, and using uncertainty to invoke the commonsense reasoning only on frames where it is needed.

**Step 3:** Use the violated rules to identify potential false predictions by assuming the detected collective behaviors are true. For example, if the collective behavior is that vehicles will change to a different lane when they are near a region of a specific lane and far away from intersections, there is only one explanation: there is an obstacle in that region. However, no obstacles were detected by the deep learning model in that region. Then this explanation is violated, so we can predict that there should be an obstacle in that region, which is a misclassification of the deep learning model.

**Step 4:** Make adjustments to the non-vehicle objects predicted by the deep learning model to make the scenario consistent again.

## 5 EXPERIMENTS

**Experimental Setup:** For each scenario, we compare the commonsense reasoning result with the baseline model output and generate a final result. The final metrics are generated by comparing the final results with the ground truth data. For traffic light scenarios, we evaluate the color of the traffic lights, and for obstacle scenarios, we evaluate the detection of the obstacle. We perform evaluations over data collected from the CARLA simulator. We use recordings from the simulator to evaluate our approach for traffic light and obstruction scenarios. In each scenario, we compare the accuracy, precision, recall, and F-measure of the hybrid model with the baseline model. Additionally, we consider two types of commonsense reasoning layers: the fully active commonsense reasoning layer and the uncertainty-invoked commonsense reasoning layer.

**Results and Evaluations:** The results of our system over the CARLA simulator data for traffic light scenarios are shown in Table 1. While the logic layer depends on the deep learning system for perception, the evaluation of our target scenarios and objects is separate. This means that for the traffic light scenario, the logic layer uses all of the data provided by the deep learning model except for the state of the traffic light. As such, we evaluate the effectiveness of both our baseline detection model and then we evaluate it when combined with the commonsense reasoning layer.

For the logic model, we only consider frames in which the logic applies, which consist of frames in which there is a nearby intersection and at least one nearby collective behavior. We also don't consider the first few frames after a light changes from red to green to give time for a collective behavior to be detected. For the baseline detection model and the combined model, we evaluate them over all frames in the dataset. In the combined model, when there is a discrepancy between the traffic light detection model and the logic model, we treat this as an inconsistency. In the system proposed in this paper, we favor the output from the logic system. In future systems, alternate actions could also be taken, such as providing a warning with an explanation, suggesting alternate action, or requesting the perception model to reevaluate a target area.

| CARLA Traffic Lights | Metrics | | | |
|---|---|---|---|---|
| | Accuracy | Precision | Recall | F-Score |
| Town 1 100 Baseline | .4789 | .6578 | .1445 | .2369 |
| Town 1 100 Combined | .8546 | .9297 | .9942 | .9608 |
| Town 1 200 Baseline | .7634 | .5 | .4091 | .45 |
| Town 1 200 Combined | .8387 | .64 | .7272 | .6809 |
| Town 2 100 Baseline | .2446 | .6757 | .0535 | .0992 |
| Town 2 100 Combined | .6672 | .8825 | .6595 | .7549 |
| Town 2 200 Baseline | .5071 | .5161 | .1524 | .2353 |
| Town 2 200 Combined | .8418 | .7939 | .9429 | .8646 |
| Town 3 100 Baseline | .4083 | .3333 | .0491 | .0855 |
| Town 3 100 Combined | .5502 | .5954 | .6319 | .6131 |
| Town 3 200 Baseline | .2573 | .8235 | .1014 | .1806 |
| Town 3 200 Combined | .8129 | .9206 | .8406 | .8788 |
| Town 4 100 Baseline | .3515 | .6774 | .0769 | .1381 |
| Town 4 100 Combined | .4059 | .7088 | .2051 | .3181 |
| Town 4 200 Baseline | .6272 | .3035 | .1197 | .1717 |
| Town 4 200 Combined | .6931 | .5412 | .3239 | .4053 |

Table 1: Results of commonsense reasoning, baseline deep learning, and combined models for traffic lights over Towns 1-4 at 100 and 200 NPC densities

## 5.1 FULLY ACTIVE COMMONSENSE REASONING LAYER

The CARLA datasets used are recordings taken over CARLA Towns 1-4 using multiple vehicle densities. We currently evaluate a total of 100 and 200 NPC vehicles (vehicles not represented by the AV system). We have also introduced inclement weather into these scenarios. These factors affect the image data and the accuracy of the traffic light detection model. As one can see in the table, the logical layer evaluates at around 70-95% for all metrics provided, while the baseline model has significantly worse performance. This provides a large increase in accuracy for the overall model when the traffic light detection model fails to achieve good accuracy. For example, in the Town 1 100 NPCs dataset, the weather condition was rainy. There were several frames in which the traffic light detection model failed to detect a red light, but this inconsistency was caught by our logic system. This allows the combined system to achieve an accuracy increase of anywhere from 5-56% over the baseline detection model in all the recordings, depending on how frequently the baseline failed on scenarios that were captured by the commonsense. In Table 2 we see the results of the scenario in which a stopped vehicle is obstructing traffic. These recordings are primarily taken from CARLA Town 3, with varying NPC densities. More dense scenarios were considered since we rely on detecting collective behaviors, which occur more frequently in more vehicle-dense environments.

In these recordings, there was an obstruction at each frame, and we evaluated whether or not the baseline model could identify the stopped vehicle. We also determined if the logic model was able to detect an obstruction by noticing a collective lane change when there were no nearby intersections. Like the traffic light scenario, we only evaluate the commonsense reasoning in frames where the reasoning applies. In these particular recordings, there was always a collective lane change, so the commonsense reasoning easily achieves 100% accuracy. The baseline model naturally struggles with identifying an obstruction on the road, since it relies on the obstruction being in camera view. It had varying performance depending on whether the obstruction was visible or being blocked by nearby traffic. In all scenarios, however, the combined model is able to reference the commonsense reasoning to achieve 100% accuracy.

## 5.2 UNCERTAINTY-INVOKED COMMONSENSE REASONING LAYER

In the following experiment, we consider the same two traffic scenarios. However, we use our uncertainty prediction models to generate an uncertainty value for the BEV semantic segmentations

| CARLA Obstacles | Metrics | | | |
|---|---|---|---|---|
| | Accuracy | Precision | Recall | F-Score |
| Town 3.0 Dense Baseline | .4943 | 1 | .4943 | .6615 |
| Town 3.0 Dense Combined | 1 | 1 | 1 | 1 |
| Town 3.1 Dense Baseline | .93 | 1 | .93 | .9636 |
| Town 3.1 Dense Combined | 1 | 1 | 1 | 1 |
| Town 3.2 Dense Baseline | .392 | 1 | .392 | .563 |
| Town 3.2 Dense Combined | 1 | 1 | 1 | 1 |
| Town 3.0 Mid Baseline | .2284 | 1 | .2284 | .3718 |
| Town 3.0 Mid Combined | 1 | 1 | 1 | 1 |

Table 2: Results of the commonsense reasoning model for obstructions for the logic, baseline, and combined models for recordings over Town 3. Dense means a larger number of NPCs in the recording while Mid means fewer.

and the camera inputs for each frame in our dataset. When the uncertainty exceeds a certain threshold, we invoke the commonsense reasoning layer to help handle the uncertainty. This is a more efficient method for allowing the AV model to deal with difficult scenarios without requiring the commonsense layer to be active at all times.

In this experiment, we apply aleatoric uncertainty towards the traffic light scenarios and epistemic uncertainty towards the obstacle scenario. For the traffic light scenarios, noise has been applied to the traffic light object detection model, causing there to be uncertainty in the quality of the traffic light classification. We took a percentage of the average uncertainty across all frames of the dataset and set it as the threshold. If an individual frame exceeded this value, then the commonsense layer activated and searched for nearby cluster behaviors to determine the status of the light. The results of this experiment can be seen in Table 3, which shows the evaluation metrics for identifying red traffic lights across four different recordings of four different towns in CARLA. The AV system on its own had decent accuracy and maintained a very high precision since it had very few false positives. However, it has a very low recall, meaning that it failed to identify several red traffic lights. The combined uncertainty/commonsense layer showed massive improvements to the results of the model. Frames that were labeled incorrectly by the baseline model often had higher aleatoric uncertainty, allowing the commonsense reasoning layer to make corrections. This severely reduced the false negative rate of the system overall. Furthermore, the uncertainty-invoked commonsense reasoning layer ran on much fewer frames than the fully active commonsense reasoning layer. Despite this, it still performed well as an optimizer for the AV system.

For the obstacle detection scenario, we changed the obstacles from stopped vehicles to animals in the road. These animals were not seen in the training set, meaning that the AV system evaluated frames that contained animals with a high epistemic uncertainty. The uncertainty-invoked commonsense reasoning layer then used this value to identify animals on the road. In this experiment, we evaluated the system's ability to detect animal objects directly in front of the ego vehicle. The animal classification was completely out-of-distribution for the baseline model, meaning that it never captured any of the animals obstacles, but it would generate a high epistemic uncertainty for frames where animals were in front of the ego vehicle. The baseline model has higher default accuracy compared to the previous obstacle experiment since the task is slightly different. In this experiment, there are very few positive labels. This emulates a real-world scenario in which an AV system will only encounter certain out-of-distribution objects very rarely, meaning that is typically accurate in the general case but will struggle handling unfamiliar objects. The uncertainty-invoked commonsense layer for this experiment ran on significantly fewer frames than the fully active commonsense layer since there were very few positive labels and, therefore, very few frames with high epistemic uncertainty. We also used a static threshold in this experiment as opposed to the dynamic threshold used in the traffic light experiment. This allows the commonsense layer to be extremely efficient since it is only expected to run when a very rare scenario occurs. Table 4 shows recordings from two different towns where the ego vehicle encountered animals and failed to identify them. The table shows that the logic model managed to create a small improvement over the baseline model by slightly improving the true positive rate.

| CARLA Traffic Lights | Metrics | | | |
| --- | --- | --- | --- | --- |
| | Accuracy | Precision | Recall | F-Score |
| Town 1.1 Baseline | .6444 | .9388 | .6133 | .7419 |
| Town 1.1 Combined | .9556 | .961 | .9867 | .9737 |
| Town 2.6 Baseline | .6136 | .9782 | .5769 | .7258 |
| Town 2.6 Combined | .8295 | .9846 | .8205 | .8951 |
| Town 3.6 Baseline | .6172 | .9565 | .423 | .5867 |
| Town 3.6 Combined | .8395 | .9756 | .7692 | .8602 |
| Town 5.6 Baseline | .7931 | 1 | .7882 | .8816 |
| Town 5.6 Combined | .9425 | 1 | .9412 | .9697 |

Table 3: Results of the baseline deep learning traffic light detection model and the uncertainty-invoked commonsense reasoning

| CARLA Obstacles | Metrics | | | |
| --- | --- | --- | --- | --- |
| | Accuracy | Precision | Recall | F-Score |
| Town 3.6 Baseline | .93 | N/A | 0 | N/A |
| Town 3.6 Combined | .9495 | 1 | .2857 | .444 |
| Town 4.5 Baseline | .94 | N/A | 0 | N/A |
| Town 4.5 Combined | .9597 | 1 | .3333 | .5 |

Table 4: Results of the baseline deep learning object detection model and the uncertainty-invoked commonsense reasoning

## 5.3 Takeaway/Discussion/Significance

The results from the experiments reveal that applying a feedback layer based on commonsense reasoning is an effective method for handling misclassifications in our perception model. In every experiment for the traffic light scenario, we saw an improvement from the baseline perception model to the combined model. The effect of external factors on the baseline perception model, such as inclement weather conditions in this case, can be heavily mitigated. A limitation of this experiment is the imperfect logic. Like we argued earlier, it is possible to achieve 100% accuracy within scenarios captured by the commonsense reasoning. However, in our experiment, the accuracy of the logic model was around 95% and reached as low as 70%. But despite this, we can still see an increase in performance from the baseline model to the combined model. This is because the commonsense reasoning module will only provide an adjustment in scenarios that are eligible for the logic. The experiment also demonstrated that uncertainty can be used to invoke commonsense reasoning in a more efficient manner while still improving the accuracy of the system.

## 6 Related Work

As research moves forward with improving autonomous vehicles, there is more and more work being done on similar hybrid artificial intelligence systems to the one proposed in this paper. Our approach features improving the knowledge extraction and visual question-answering aspects of autonomous vehicles using collective behaviors; however, similar work has been done recently.

For example, the work done by Kanapram et al. (Kanapram et al. (2020)) explores collective awareness in a network of agents. They explore how collective behaviors can be detected in a connected system of autonomous vehicles. These collective behaviors are modeled using Dynamic Bayesian Networks and used to identify abnormal situations on the road. This approach uses a statistical relational approach as opposed to a logical approach to identify abnormal scenarios, but it demonstrates a similar concept in using collective behaviors to handle unusual road scenarios. Both approaches are able to generate information about future states based on current information about our scenario. Our approach, however, more closely models human reasoning and can provide a more readable explanation of the scenario, as we are not relying on conditional distributions to identify abnormalities. We also solely rely on camera data to detect abnormal scenarios, as opposed to needing information

from a group of agents. This does come at the cost of scalability. This sort of agent based approach to identifying collective behavior is also explored by Schwarting et al. (Schwarting et al. (2019)). They designed a system that measures Social Value Orientation to predict the behaviors of vehicles on the road. Their approach is similar to ours in that they use the behaviors of vehicles to generate new information about the current road scenario. We agree with their assertion that using the behaviors of other vehicles in an AV system allows it to be more intelligent and socially aware.

Research into more logical hybrid systems has also been explored in recent years. Bannour et al. (Bannour et al. (2021)) propose a symbolic model-based approach to generating logical scenarios for autonomous vehicles. Suchan et al. have also explored many models (Suchan et al. (2020; 2021)) that combine commonsense reasoning with AVs to create more human-centred, explainable, and ethical vehicles. Their work more closely explores general applications of ASP-based hybrid perception systems, while ours demonstrates a specific commonsense-based optimization. The advantage of our approach is that we can apply our technique to existing AV systems since the commonsense reasoning is decoupled from the perception. Another work, called AUTO-DISCERN, by Kothawade et al. (Kothawade et al. (2021)) proposes a framework that automates AV decision-making using answer set programming (ASP). Their work focuses on handling the decision-making process using ASP, while our approach uses commonsense reasoning as a method of *improving* an AV system. In this sense, our approach is more viable for real-world applications for AV systems. With our approach, we can generate a commonsense base for important safety-critical scenarios and always show an improvement in the decisions created by deep learning-based AVs, as opposed to needing to generate and invoke a large commonsense reasoning base that represents all possible driving decisions and scenarios. This is further emphasized by the fact that AUTO-DISCERN was also not evaluated over any simulators, as we did with CARLA, and their evaluations were ad hoc.

There is significant research on quantifying these uncertainties in DL models, typically designed for i.i.d. inputs like images and tabular data. Many existing methods leverage the concept of Bayesian model averaging overall multiple forwrad passes to estimate distributions over weights or multiple models, such as deep ensembles (Lakshminarayanan et al. (2017)) and dropout-based Bayesian neural networks (BNNs) (Gal & Ghahramani (2016)). However, the requirement of excessive memory and computation burden for training and testing makes it impossible for real-time applications. Recently, a new class of models has been proposed based on the concept of Evidential Deep Learning Ulmer et al. (2023). They allow uncertainty quantification in a forward pass and with a single set of weights by parameterizing the distributions over distributions.

## 7    CONCLUSION AND FUTURE WORK

We assert that an accurate logic model serves as an effective method for improving the accuracy of autonomous vehicle models through consistency checking. This is indeed demonstrated by our experiments. Because the layers are well separated, we can easily adjust the logical layer to maintain high accuracy without touching any black box models that it is layered on top of. We additionally demonstrated that uncertainty can be used to more efficiently invoke commonsense reasoning. As such, our system reveals an adjustable, explainable, and effective way to perform optimizations for autonomous vehicles. Our future research will focus on building a scalable knowledge base to cover the entire class of possible misclassifications for an autonomous vehicle. We also plan to evaluate the effectiveness of our approach on real-world datasets and systems.

Additionally, a desirable extension to this work is to use Answer-Set Programming (ASP) (Gelfond & Kahl (2014)) to create a more sophisticated modeling of the commonsense knowledge explored in this paper. Commonsense reasoning is heavily based on human reasoning techniques such as defaults, constraints, exceptions, abduction, etc., (Gelfond & Kahl (2014)) which can be represented more precisely using ASP. Our future research will involve modeling different possible worlds based on missing information in perception and generating more thorough explanations for a misclassification. Our ultimate goal is to build an AV system that layers commonsense reasoning on top of perception to achieve 100% accuracy resulting in SAE Level 5 autonomy. The research reported in this paper proves that this is indeed possible. Thus, our main contribution is to demonstrate that perception models augmented with commonsense reasoning can lead to SAE Level 5 autonomy.

REPRODUCIBILITY STATEMENT

To facilitate the reproducibility of the results in this experiment, we have provided the source code for recording data from CARLA, the code to convert the recorded data into facts, the commonsense reasoning knowledge base used, and the code to generate the evaluations. These components are used to perform the general procedure outlined at the end of Section 4. Furthermore, the experimental setup described at the start of Section 5 outlines how the evaluations are performed for the given programs.

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
