# OpenReview forum: "Towards Fully Autonomous Driving with Automated Commonsense Reasoning"
_ICLR.cc/2025/Conference — Submitted to ICLR 2025_

### Official Review · Reviewer_qgQR · 2024-10-22

**Soundness:** 1
**Presentation:** 2
**Contribution:** 1
**Rating:** 1
**Confidence:** 3

**Summary:**

The authors apply a set of hand-crafted, logic rules to a BEV segmantic map. In case the rules are not consistent with the segmantic map, a correction is triggered, that makes the scene consistent again.The approach is evaluated in two different scenarios, using various CARLA towns. The authors also analyze the effect of using uncertainty as a metric to determine if the rule correction is possible relevant.

**Strengths:**

Handling of corner cases, as considered by the authors, is a long standing problem in autonomous driving. Till now, no generalized solution is available and rule-based approaches offer an interesting path, as they are interpretable, as opposed to common (black box) deep learning models. Therefore, additional insight into this class of methods is significant. Textually, it is clearly written but lacks structurally (see weaknesses).

**Weaknesses:**

The most substantial weakness concerns details wrt. the method and the experiments. In general, it should be state which factoids can be used and if the rule system allows for temporal logic. A flowchart and/or example would also greatly help to understand the details of the method.

Concerning the experiments, it needs to be clarified how the used rules look like, what exactly the prediction (evaluation) target is and how parameters (like thresholds) are selected. Furthermore, the experiments are currently not reproducible, because additional information is also missing. As example, how the weather variations are defined and how to interpret the "number of cars" (cars per ... ?)
Furthermore, it seems the experiments are designed to align with experiment setup, meaning that the used rules and uncertainty type are always correct. E.g. what happens when a convoy exits a highway? Additional studies are also required to be able to judge the results in a meaningful way: A comparison to baselines (like the ones state in related work) and relations between the number of other vehicles and the effectiveness of the methods (required levels of local density & consensus). Additionally, the evaluation seems to be performed on perception level, but it should be performed on policy level, because the stated issues can be resolved with policies, without a correct perception.

Another problem concerns hidden assumptions: The rules are assumed to be perfect, which is likely not possible in reality. Additionally, the authors assume that it is possible to hand-craft a sufficient number of rules to cover most/all relevant corner cases. This is also not viable. Of course, there are ways to possibly automatize but this is not discussed and a comparison with existing, automatized methods is also missing.

These issues also reoccur in some claims, that need to be substantiated: This approach can likely not lead to 100% accuracy, because there is not even 100% agreement between expert drivers. Why is high level reasoning/system 2 required? End-2-End stacks are arguably not working like this. Why must the system comprehend and reason like a human would?

Lastly, the structure of the paper makes it difficult to follow: The methology sections contains experiment settings, related work is only discussed in the very end. Background and Introduction are mixed. Furthermore, some references are missing (e.g. resnext50)

**Questions:**

1. The deep learning model outputs a large amount of image and object data to represent each road scenario. Why do they output image data?
2. A script is used to convert the data from these objects into facts that can be used by the commonsense reasoning module. - Is this format conversion, or does it contain logic (with assumptions)?
3. We currently evaluate a total of 100 and 200 NPC vehicles - total per what?
4. We have also introduced inclement weather into these scenarios - What exactly does this mean?
5. In these recordings, there was an obstruction at each frame How can there be an obstruction in each frame? Only according frames selected?
6. What does "dynamic threshold" mean?

---

### Official Review · Reviewer_Tf9v · 2024-11-01

**Soundness:** 1
**Presentation:** 2
**Contribution:** 1
**Rating:** 3
**Confidence:** 4

**Summary:**

The paper introduces a novel approach to achieving fully autonomous driving by combining deep learning with commonsense reasoning.

It proposes a hybrid architecture with a base layer using deep learning for environmental perception and target detection, and an upper layer using “commonsense reasoning” to identify and correct misclassifications made by the deep learning models.

The effectiveness of this approach is demonstrated in scenarios： 1. traffic light failure and 2. unexpected obstacles on the road.

The paper claims that by integrating commonsense reasoning with deep learning, the robustness and safety of AD systems can be significantly enhanced. Authors claims they offer a promising path towards achieving fully autonomous driving.

**Strengths:**

I do believe that it would be beneficial for the AD system to incorporate elements of human common sense and reasoning.

The paper provides a way for efficiently invoking commonsense reasoning by measuring uncertainty in the computer vision model and using commonsense reasoning to handle uncertain scenarios.
While individual components in the paper are not novel, the attempt to bridge the gap between deep learning approaches and human reasoning capabilities is interesting.

The implementation on the CARLA simulator enhances reproducibility, further supported by the authors' commitment to open science through their detailed documentation and open-sourced code.

**Weaknesses:**

- The motivationof the paper seems not convincing. The example in Figure 1, which highlights errors in traffic light recognition, seems outdated. Current state-of-the-art end-to-end systems have largely addressed this kind of issue with high accuracy. The authors should focus on more prevaling challenges within AD, such as complex urban scenarios or interactions with non-standard road users.

- The method of observing surrounding vehicles to infer basic information is not novel. Many existing AD methods already incorporate similar strategies for behavior prediction. The paper should clarify how their approach extends beyond observational methods to real "commonsense" reasoning.

- The experiment section lacks clarity, particularly regarding the baseline model used. The paper should provide detailed descriptions of the baseline architecture, training procedures, and performance metrics. Additionally, the integration mechanism between the commonsense reasoning layer and the baseline detection model needs to be clearly explained, including how these components interact and resolve potential conflicts.

- Besides, the paper lacks a comprehensive literature review. Given the rapid advancements and extensive research in this field, a more up-to-date thorough engagement with existing literature is necessary.

**Questions:**

See weakness part, please address my concerns.

---

### Official Review · Reviewer_XAox · 2024-11-02

**Soundness:** 3
**Presentation:** 2
**Contribution:** 3
**Rating:** 5
**Confidence:** 3

**Summary:**

This paper proposes a commonsense reasoning layer to advance the autonomy of self-driving vehicles, aiming for SAE Level 5 autonomy. It addresses limitations in machine learning-based models by incorporating commonsense reasoning to handle ambiguous scenarios. Two primary cases are explored:

 - Malfunctioning traffic lights at intersections.
 - Unexpected obstacles on the road.

The authors demonstrate that combining deep learning with commonsense reasoning improves decision-making accuracy in the CARLA simulator environment. Uncertainty quantification activates the commonsense layer only when needed, optimizing computational efficiency for real-time systems. This approach showcases how human-like reasoning, structured as a logical model atop a deep learning model, enhances the accuracy of perception tasks in self-driving vehicles.

**Strengths:**

The paper presents an approach for self-driving vehicles to interpret and respond to complex situations by integrating commonsense reasoning, which mimics human-like decision-making processes. Additionally, the scenarios where this approach is applicable are well explained, helping readers easily understand how commonsense reasoning can enhance decision-making in ambiguous or novel driving situations.

- The uncertainty-based approach is efficiently applied, simultaneously improving computational efficiency and decision-making accuracy, both vital for real-time applications. By activating the commonsense layer only in high-uncertainty situations, the method optimizes computational resources, which is essential for system deployment.
- The paper extends its application beyond the initial traffic light recognition and obstacle detection scenarios, suggesting that the commonsense reasoning approach could generalize to handle diverse driving situations.
- The choice of Evidential Deep Learning (EDL) for uncertainty quantification is well-motivated by computational cost considerations. EDL is advantageous in scenarios where real-time efficiency is critical, as it allows uncertainty estimation in a single forward pass without requiring multiple model evaluations.

**Weaknesses:**

The lack of explicit details on the logic programming rules makes it difficult to verify or reproduce the commonsense reasoning layer’s effectiveness, posing a challenge for reproducibility. Additionally, clear architecture diagrams are missing, making it harder to visualize the system’s workflow and understand how each component interacts.

- Rule-based reasoning generally lacks flexibility in handling unexpected and highly complex situations, as fixed rules can be rigid and challenging to generalize for unpredicted events.
- The use of Uncertainty Quantification primarily to invoke the commonsense layer, rather than directly contributing to the reasoning process, somewhat limits the integration depth of UQ with the logic layer.
- **The assumption that commonsense reasoning reflects human-like decision-making may not hold in scenarios where all vehicles in the environment are autonomous, raising questions about the relevance of “commonsense” when collective behaviors are not human-driven.**

**Questions:**

**Q1.** In situations where all surrounding vehicles are autonomous (as could be expected in future Level 5 autonomy scenarios), do you believe that commonsense reasoning based on collective human-like behavior remains relevant? How would this framework adapt if collective actions are not based on human drivers?

**Q2.** The paper specifies that aleatoric uncertainty is used for the traffic light scenario, and epistemic uncertainty is used for obstacle detection. However, it would be helpful to explicitly define which quantified metrics from Evidential Deep Learning (EDL) correspond to aleatoric and epistemic uncertainty. Specifically, detailing how the variance within the dirichlet distribution indicates aleatoric uncertainty, while the concentration or overall confidence level of the dirichlet distribution corresponds to epistemic uncertainty, would provide clarity. This explicit mapping would strengthen the understanding of how EDL’s quantified outputs are applied to each uncertainty type in these scenarios. Could you specify which exact metrics from dirichlet distribution in EDL are mapped to aleatoric and epistemic uncertainty in each scenario?

---

### Official Review · Reviewer_RXLu · 2024-11-04

**Soundness:** 2
**Presentation:** 1
**Contribution:** 1
**Rating:** 1
**Confidence:** 3

**Summary:**

The paper attributes the lack of SAE Level 5 autonomous vehicles on the market primarily to an over-reliance on machine learning. It proposes incorporating automated commonsense reasoning using hand-coded logic programs alongside existing ML models, as a solution. To demonstrate the benefits, the paper presents two scenarios using the CARLA simulator.

**Strengths:**

The paper highlights alternative approaches to address the challenges of out-of-distribution data in machine learning models. It conducts tests and analyses on standard evaluation settings and provides source code for the experiments as supplementary material.

**Weaknesses:**

* **W1:** While the methodology of using logic programs to safeguard against unsafe decisions is noteworthy, similar approaches, such as shielding mechanisms to safeguard against unsafe actions, have been central to planning under uncertainty in works published around 2015–2020 (see references below)

* **W2:**  There are several approaches to applying commonsense reasoning. A significant limitation of logic program-based approaches has been their restricted generalization to unseen scenarios. As a result, the question of how these methods can be effectively used for plausibility testing or achieving system self-awareness remains open. The paper does not offer a new perspective on this issue. Recently, using Large Language Models has emerged as a promising approach (see reference below).

* **W3:** In the experiments section, comparisons with additional state-of-the-art approaches would enhance the evaluation, rather than relying on a single baseline.

* **W4:** The presentation style could be enhanced to increase information density, particularly in Section 3, "Motivation." Textual comparisons with human behavior could be replaced with logical formulas. Additionally, explanations about BEV (Lines 208-213) can be omitted. Details on CARLA’s sensor configurations, parameter values, and API commands could be moved to the appendix, as they do not contribute methodologically.

* **W5:** The "General Procedure" section could be specified using variables and mathematical notation, which would facilitate analysis of bounds and enable a clearer abstraction of the example scenarios. The textual description could also shift from a reporting style to an analysis that examines causal relationships between design choices and outcomes. Additionally, the sentences in lines 503-506 could be revised for improved flow.

* **W6:** The related work section may benefit from organization into subcategories, such as approaches for commonsense reasoning, uncertainty quantification, and self-awareness. Capitalization rules should be observed in the bibliography entries.

1. Shalev-Shwartz et al. "On a formal model of safe and scalable self-driving cars", 2017.
2. Tas et al. "Automated vehicle system architecture with performance assessment", 2017
3. Hess et al. "Formal Verification of Maneuver Automata  for Parameterized Motion Primitives", 2014
3. Alshiekh et al. "Safe Reinforcement Learning via Shielding", 2018
5. Sinha et al. "Real-Time Anomaly Detection and Reactive Planning with Large Language Models", 2024

**Questions:**

* **Q1:** In Line 318: "_We also don’t consider the first few frames after a light changes from red to green to give time for a collective behavior to be detected._".  Does this introduce a dead time into the system? How many frames are omitted?

* **Q2:** Since weather conditions are varied in your experiments, would it be beneficial to include explanations of the weather conditions in Table I?

---

### Meta-Review · Area_Chair_m7zy · 2024-12-18

**Metareview:**

Reviewers unanimously had many strong concerns about this paper and none rated it above a 5, and two reviewers rating it a 1. The authors did not respond to any concerns. Among the variety of issues raised by the reviewers, these are the particularly salient issues:

- The paper is similar to prior work, resulting in an unclear contribution. Several reviewers pointed out similarity to prior work and an unconvincing application to traffic light recognition
- The above issue is related to issues about lacking a comprehensive literature review
- The paper is missing important details about the method, making it difficult to reproduce
- The paper is missing discussion about the appropriateness of the assumption that rules are 100% accurate
- The paper includes comparison to only a single baseline (https://openreview.net/forum?id=V1N6MmDY27&noteId=8gMgF79Pfu), and important details about it are missing

These issues are part of a unanimously negative evaluation. A brief read of the paper has also led me to concur with the negative evaluation.

**Additional Comments On Reviewer Discussion:**

No discussion occurred, because the authors didn't respond.

---

### Decision · Program_Chairs · 2025-01-22

Reject